# Effect of Excessive Screen Time on Cardiorespiratory Fitness in Children: A Longitudinal Study

**DOI:** 10.3390/children9101422

**Published:** 2022-09-20

**Authors:** Ryo Goto, Tsunenori Isa, Rika Kawaharada, Kana Horibe, Yamato Tsuboi, Kiyomasa Nakatsuka, Kazuaki Uchida, Kenta Saeki, Rei Ono

**Affiliations:** 1Graduate School of Health Sciences, Kobe University, 7-10-2 Tomogaoka, Suma, Kobe 654-0142, Hyogo, Japan; 2National Institutes of Biomedical Innovation, Health and Nutrition, Department of Physical Activity Research, Toyama 1-23-1, Shinjyuku, Tokyo 162-8636, Japan

**Keywords:** sedentary behavior, mobile phone, physical fitness, child development

## Abstract

Background: This study investigated the effect of different components of screen time (mobile phone use, TV/video viewing, and video gaming) on cardiorespiratory fitness (CRF) development in children aged 9–12 years. Methods: This was a two-year longitudinal study conducted with 175 children (49.7% girls, mean age = 9.5) in Japan. CRF was assessed using a 20 m shuttle run test conducted at baseline and again at follow-up. Children were categorized as “Good” or “Poor” based on the change in CRF scores for each gender. Screen time was assessed using a self-reported questionnaire at baseline and termed as “high” if children reported ≥ 2 h/day. Univariate and multivariate logistic regression analyses were performed after adjusting for gender, physical activity, and time of data collection. Results: Children scoring “high” on mobile phone use had lower odds of being categorized as “Good” in CRF change (crude odds ratio (OR): 0.34; 95% confidence interval (CI): 0.15–0.90 (adjusted OR: 0.33; 95% CI: 0.12–0.91)). There were no significant effects of TV/video viewing (crude OR: 1.54; 95% CI: 0.84–2.81) and video gaming (crude OR: 0.98; 95% CI: 0.48–1.97) on changes in CRF. Conclusions: Limiting excessive mobile phone usage might be important for ensuring healthy development of CRF in children.

## 1. Introduction

There has been a substantial decline in cardiorespiratory fitness (CRF) among children in developed countries since the 1980s [1,2,3]. In children, lower CRF is associated with poor academic achievement, adiposity, and onset of depression and anxiety [4,5,6,7]. On the other hand, CRF is associated with preventing the negative health effects of stress in youth and improving social health [8,9]. Thus, it is important to understand the cause behind the substantial decline in CRF.

The increasing amount of time spent in front of screens has been identified as one cause for this [10]. Previous studies have reported that the total screen time, including watching TV/videos, playing video games, and using a computer, was negatively associated with changes in CRF in children between the ages of 11 to 13 [11,12]. In addition to this, the percentage of children aged 8–11 years who own a mobile phone has been increasing over the years (from 13% in 2010 to 35% in 2017), along with an increase in mobile phone use for watching videos and playing games and a decrease in time spent watching TV [13]. Consequently, a study in 2015 reported that mobile phones had been added to the total screen time, and this total screen time was also negatively associated with CRF [14]. Therefore, it is necessary to investigate the effect of mobile phone use, in addition to TV/video viewing and video gaming, on CRF development.

However, to the best of our knowledge, there have been only a few studies that have investigated the longitudinal association between mobile phone use and CRF development. Similarly, there have been a few studies that have investigated the difference in the effect of the various components of screen time (TV/video viewing, video gaming, and mobile phone use) on CRF development. Furthermore, it is important to investigate the relationships between screen time and CRF during the age when maximum development of CRF takes place (for boys, at 11.4 years; for girls, at 10.5 years) [15].

In line with this, the purpose of this study was to investigate the effect of each type of screen time (including mobile phone use, TV/video viewing, and video gaming) on CRF development in children between the ages of 9–10 to 11–12 years.

## 2. Methods

### 2.1. Participants

This was a 2-year longitudinal study. We recruited fourth-grade children from two public elementary schools in Kobe, Japan, as participants from May to October 2016 and from May to October 2017. The inclusion criteria were those who gave consent to participate in our study. The exclusion criteria were as follows: (1) children who missed follow-up and (2) children whose data were missing. We explained the study protocol to the principals, teachers, and children and obtained informed consent from both the children and their guardians. This study protocol was approved by the Research Ethics Committee of Kobe University Graduate School of Health Science (Approval number: 545-3).

### 2.2. Procedure

A self-reported questionnaire, anthropometric measurements, and a 20 m shuttle run test were used to gather data at both baseline (2016 and 2017) and at the 2-year follow-up (2018 and 2019). The children answered the questionnaire with support from well-trained research assistants. Anthropometric measurements and 20 m shuttle run test were conducted by elementary school teachers and the school staff.

### 2.3. Measurements

(1)Demographic data

The children’s age and gender were ascertained from the self-reported questionnaire. Their height (cm) and weight (kg) were obtained from the records of the routine health examinations conducted at the elementary schools. Body mass index (BMI) was calculated as weight in kilograms divided by height in meter squared (kg/m^2^).

(2)Cardiorespiratory Fitness (CRF)

CRF was assessed using the 20 m shuttle run test, which is a valid and reliable way of assessing CRF in children [16]. The 20 m shuttle run test was conducted according to the guidelines for physical fitness measurement prescribed by the Ministry of Education, Culture, Sports, Science, and Technology of Japan [17]. Participants ran back and forth between lines set 20 m apart while keeping pace with audio signals emitted from a pre-recorded compact disc. The initial speed was 8.5 km/h, and the pace was increased by 0.5 km/h every minute. The test ended when the participants could no longer maintain the pace or could not complete two laps continuously within the time limit. The total number of laps was recorded at the end of the test.

(3)Screen time

Information regarding screen time was obtained using a self-reported questionnaire. We assessed screen time spent on each electronic device, including mobile phone use, TV/video viewing, and video gaming, during the daytime. Participants indicated their daily screen time as “no time spent”, “30 min”, “1 h”, “2 h”, “3 h”, “4 h”, “5 h”, “6 h”, or “≥7 h” [18,19]. From 2017 to 2019, we assessed the average time spent daily during the daytime on each component. In 2016, we assessed the time spent on weekdays and weekends separately and calculated the daily average time as follows: {(weekday times) × 5 + (weekend time) × 2}/7. There is a moderate-to-high agreement between repeated responses to the screen time questions, with Kappa coefficients ranging from 0.68 to 0.82 [20]. We categorized the duration of each component of screen time into <2 h/day and ≥2 h/day [21,22].

(4)Physical Activity (PA)

We assessed PA using the Japanese version of the Physical Activity Questionnaire for Older Children (PAQ-C), which has acceptable reliability and validity (Cronbach’s alpha = 0.81, intraclass correlation coefficients = 0.83) [23]. The PAQ-C is a self-reported 7-day recall questionnaire that measures general levels of moderate to vigorous PA in 8–14-year-olds, during the school year. The PAQ-C consists of nine computable items, and each item was scored on a 5-point scale, with higher scores indicating higher PA levels. The total PAQ-C score was calculated as the mean score of the nine items, ranged between 1.00 (lowest activity level) and 5.00 (highest activity level), and was used as a continuous variable [24].

### 2.4. Statistical Analysis

Within each gender group, children were divided into tertiles on the basis of their performance in the 20 m shuttle run test at baseline and follow-up. Children were categorized as “Poor” if they moved down a lower tertile during follow-up or if they maintained the lowest tertile. Alternately, children were categorized as “Good” if they moved up to a higher tertile or if they maintained the middle or the highest tertile. A univariate logistic regression analysis was used to assess the effects of different components of screen time at baseline on the changes observed in CRF over the two-year period. The analysis was performed using screen time as an independent variable and the change in CRF as the outcome variable (1: Good and 0: Poor). Multivariate logistic regression analysis was also performed while adjusting for gender, physical activity, and time of data collection (2016 or 2017), since screen time and CRF have been found to be associated with gender and PA [10,25]. The level of significance was set at *p* < 0.05. The data were analyzed using the free software EZR Version 1.40 (Jichi Medical University Saitama Medical Center, Saitama, Japan) [26]. 

## 3. Results

Figure 1 reveals the participant flow during the study. We recruited 203 children, and all 203 were initially included. However, we then excluded 14 children who had missed follow-up and another 14 whose data were missing (CRF, n = 14). Finally, 175 children were included in the analysis (mean age ± standard deviation: 9.5 ± 0.5; boys, n = 88 and girls, n = 87).

Table 1 presents the study sample characteristics according to time point (baseline and follow-up) and gender. During the follow-up period, mean laps in the 20 m shuttle run test increased by 14.4 for boys and by 12.9 for girls. A total of 99 children were categorized as “Good” based on the changes in their scores for CRF. At the baseline, 23 (13%) children were using mobile phones, 98 (56%) were watching TV/videos, and 41 (23%) were playing video games for two hours or more a day. Further, among these children who spent 2 h or more a day on their screens, the number of children who were categorized as “Good”, with respect to their 2-year changes in CRF, was 8/23 (35%) for those using mobile phones, 60/98 (61%) for those watching TV/videos, and 23/41 (56%) for those playing video games.

Table 2 shows the results of the logistic regression analysis. In the univariate logistic regression models, children who used their mobile phones for 2 h or more a day had significantly lower odds of being categorized as “Good” based on the changes in their CRF scores over the two years, as compared with those who used their phones for less than 2 h a day (odds ratio: 0.34; 95% confidence interval: 0.15–0.90). Similarly, in the multivariate logistic regression models, children who used their mobile phones for 2 h or more a day also had lower odds of being categorized as “Good” based on the changes in their CRF scores, as compared with those who used their phones for less than 2 h a day (odds ratio: 0.33; 95% confidence interval: 0.12–0.91). There were no significant effects found for TV/video viewing (odds ratio: 1.54; 95% confidence interval: 0.84–2.81) or video gaming (odds ratio: 0.98; 95% confidence interval: 0.48–1.97) on the change in CRF.

## 4. Discussion

This study investigated the longitudinal association between different components of screen time (mobile phone use, TV/video viewing, and video gaming) and changes in CRF over a period of 2 years among 9–12 year old children. We found that children who used mobile phones for ≥2 h/day had poor CRF development, independent of their gender, PA, and time at which the data were collected. However, TV/video viewing and video gaming were not associated with changes in CRF over the two-year period.

Previous studies have shown that an increase in total screen time negatively impacts CRF development in children [11,12]. Our results supported the findings from previous studies. Furthermore, the present study was the first to investigate the longitudinal association between different components of screen time and CRF.

Excessive sedentary behavior might be one explanation as to why mobile phone use changed CRF over a two-year period. Screen time accounts for an overall sedentary time among children and youth [27]. Additionally, excessive sedentary behavior influences lower CRF [28]. Elementary school children mainly use mobile phone for playing games [29]; online video viewing and playing video games time on mobile phones were increasing, while TV viewing time was decreasing [14]. Since mobile phones have become widespread and is replacing TV and video games [13], even children using mobile phone only, may present excessive sedentary behavior. Therefore, it was possible that ≥2 h/day mobile phone use negatively affected CRF development. However, future studies are needed to determine the difference in the effect of each device usage on CRF development since our samples who were ≥2 h/day for TV/videos and video games were larger than for mobile phones.

Another possible explanation for the negative effect of mobile phone use on CRF development could be unhealthy sleeping habits. Inadequate amounts of sleep and poor sleep quality have previously been associated with poorer CRF development [30,31]. There are several ways in which mobile phone use can cause difficulties in sleep. First, due to the pulse-modulated radiofrequency fields used in mobile phones, greater use of mobile phones has been associated with lower sleep duration [32]. Furthermore, prolonged use of wireless phones is associated with lower concentrations of β-trace protein (lipocalin-type prostaglandin D synthase), which is a key enzyme in the synthesis of prostaglandin D2, an endogenous sleep-promoting neurohormone [33]. Second, nocturnal awakening due to incoming text messages and calls leads to poorer sleep quality, including problems with falling asleep and disturbed sleep [34]. Third, due to the portability of mobile phones and ease of connectivity, the tendency to develop an addiction to them is stronger than that for a TV [35]. Greater mobile phone addiction has also been found to increase the risk of poor sleep quality [36]. In view of these reasons, it seems likely that excessive mobile phone use might lead to unhealthy sleeping habits, which in turn negatively affect CRF development; although, this study did not investigate the purpose of mobile phone use and sleep habits. Future studies need to include the purpose of screen time and sleep habits, clarifying the factor that influences CRF.

While our results were consistent with the findings from previous research [11,12], this study had several limitations. First, the sample size was small. Second, screen time and physical activity were assessed using self-reported questionnaires. Although these questionnaires were evaluated in terms of their reliability and validity [20,24], more objective measures of screen time and physical activity can be considered for use in future studies. Finally, this study lacked maturity status and parental economic status data, which could have influenced the findings. These variables should be considered in future longitudinal studies.

## 5. Conclusions

This study revealed that mobile phone use had a negative effect on CRF development over a period of two-year period, among 9–12 year old children. This suggests that, in today’s world, where several types of electronic devices exist, limiting excessive mobile phone use might be important for ensuring the healthy development of CRF in school children.

## Figures and Tables

**Figure 1 children-09-01422-f001:**
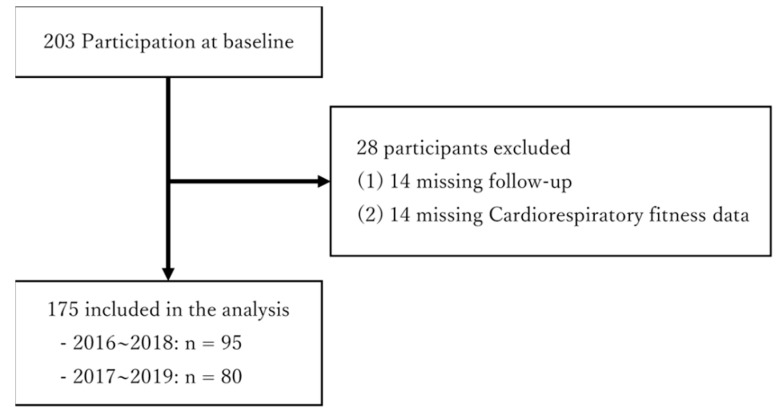
Flow chart of participation, detailing loss to follow-up and final participant sample used in this study (n = 175).

**Table 1 children-09-01422-t001:** Characteristic of the study sample (n = 175).

	Baseline						Follow-Up					
	Boy n = 88			Girl n = 87			Boy n = 88			Girl n = 87		
**Age**	9.5	±	0.5	9.5	±	0.5	11.5	±	0.5	11.5	±	0.5
BMI (kg/m^2^)	16.7	±	2.2	16.8	±	2.1	18.1	±	2.8	18.3	±	2.5
CRF (20 m shuttle run test)	42.1	±	18.4	28.3	±	12.1	56.5	±	19.2	41.2	±	14.9
CRF tertile												
The highest tertile	64.1	±	10.4	44.2	±	8.3	77.1	±	10.1	58.9	±	7.4
The middle tertile	40.6	±	3.5	27.4	±	4.1	58.4	±	4.3	39.4	±	4.8
The lowest tertile	23.2	±	8.0	16.6	±	3.3	37.1	±	10.9	25.4	±	4.2
PAQ-C	3.06	±	0.78	2.60	±	0.75	2.78	±	0.70 ^a^	2.31	±	0.58 ^b^
Screen time (≧2 h/day)												
Mobile phone use	14		(16)	9		(10)	27		(32) ^c^	38		(42) ^d^
TV/videos viewing	50		(57)	48		(55)	60		(71) ^c^	53		(62) ^d^
Video gaming	26		(30)	15		(17)	47		(55) ^c^	22		(26) ^d^

*Note*: Data show mean ± SD or n (%). BMI = body mass index; CRF = cardiorespiratory fitness; PAQ-C = Physical Activity Questionnaire for Older Children. The number of missing data: ^a^ 7,^b^ 3,^c^ 3,^d^ 2.

**Table 2 children-09-01422-t002:** Univariate and multivariate logistic models assessing the effects of each screen time on 2 years change in CRF.

		Univariate		Multivariate	
		OR	95% CI	OR	95% CI
Mobile phone use	<2 h	1.00	-	1.00	-
	≥2 h	0.34	0.15–0.90 *	0.33	0.12–0.91 *
TV/videos viewing	<2 h	1.00	-	-	-
	≥2 h	1.54	0.84–2.81	-	-
Video gaming	<2 h	1.00	-	-	-
	≥2 h	0.98	0.48–1.97	-	-

*Note:* *: *p* < 0.05. CRF: cardiorespiratory fitness. OR: odd ratio. CI: confidence interval. In multivariate analysis, adjusting for gender, physical activity, and time of data collection.

## Data Availability

The data are available upon request to the corresponding author.

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
