# Peer review of "Effect of Excessive Screen Time on Cardiorespiratory Fitness in Children: A Longitudinal Study"

_children, 2022, doi:10.3390/children9101422_

Round 1

Reviewer 1 Report

Title: Effect of Excessive screen time on Cardiorespiratory Fitness in Children: A Longitudinal Study

Article Type: Article

Summary

In this longitudinal study, the authors investigated the effect of different components of screen time on cardiorespiratory fitness development in children. Participants were 175 children (both gender) from Japan. Cardiorespiratory fitness was assessed using a 20 m shuttle run test conducted at baseline and again at follow-up. According to the baseline assessment of Screen time by a self-reported questionnaire, participants termed as “high” if they reported ≥ 2h/day. The results indicated that only children with high on mobile phone but not TV/video viewing and video gaming use had lower (poor) Cardiorespiratory fitness.

Minor points and suggestions

-          How the authors calculated sample size?

-          Please add inclusion criteria to the method section.

-          In participants section, please speak about sample, size, average age, etc.

-          How the authors calculated the cut point for good and poor categorization of Cardiorespiratory fitness?

-          How the authors did control motor development process and even social economics situation of children during these two years?

Reviewer 2 Report

Congratulations to the articles for the study.

The work is methodologically sound and well written.

Some clarifications are recommended.

In the abstract it states that the longitudinal study is over 2 years but in the procedure it speaks in lines 65-66 of "...both baseline (2016 and 2017) and at the 2-year fol- 65 low-up (2018 and 2019)." However on line 90 it talks about From 2017 to 2019. Please clarify this question.

Why was BMI not used in the multivariate analysis as an adjustment variable?

Lines 28-30, it is recommended to expand on the benefits that the CRF has for young people such as https://doi.org/10.1371/journal.pone.0259087 
